# Pyrotinib Targeted EGFR-STAT3/CD24 Loop-Mediated Cell Viability in TSC

**DOI:** 10.3390/cells11193064

**Published:** 2022-09-29

**Authors:** Xiao Han, Yupeng Zhang, Yin Li, Zhoujun Lin, Xiaolin Pei, Ya Feng, Juan Yang, Fei Li, Tianjiao Li, Zhenkun Fu, Changjun Wang, Chenggang Li

**Affiliations:** 1State Key Laboratory of Medicinal Chemical Biology and College of Pharmacy, Nankai University, Tianjin 300350, China; 2Heilongjiang Provincial Key Laboratory for Infection and Immunity, Department of Immunology, Wu Lien-Teh Institute, Heilongjiang Academy of Medical Science, Harbin Medical University, Harbin 150081, China; 3Department of Breast Surgery, Peking Union Medical College Hospital, Beijing 100730, China

**Keywords:** TSC2^−/−^, pyrotinib, EGFR/STAT3 signaling, CD24

## Abstract

Pyrotinib is an irreversible pan-ErbB receptor tyrosine kinase inhibitor, designed for the therapy of HER2-positive breast cancers. Inhibition of the epidermal growth factor receptor (EGFR, HER family) efficiently and selectively suppresses the proliferation of human TSC2-deficient smooth muscle cells and reverses lung changes in LAM/TSC. Our pilot study indicated that pyrotinib dramatically restrained the vitality of TSC2-deficient cells compared to its limited impact on TSC2-expression cells. Pyrotinib induced G1-phase arrest and triggered apoptosis by blocking abnormally activated CD24 in TSC2-deficient cells. CD24 is not only an important immune checkpoint, but is also involved in the regulation of signaling pathways. Pyrotinib inhibited the nuclear import of pEGFR and restrained the pEGFR/pSTAT3 signals, which directly boosted the transcriptional expression of CD24 by binding to its promoter region. In reverse, CD24 enhanced pEGFR function by directly binding. Pyrotinib specifically targeted TSC2-deficient cells, inhibited tumor cell viability and induced apoptosis through EGFR-STAT3/CD24 Loop in vivo and in vitro. Thus, pyrotinib may be a promising new therapeutic drug for TSC treatment.

## 1. Introduction

Tuberous sclerosis complex (TSC) exhibits aberrant activation of the mechanistic target of rapamycin complex1 (mTORC1) due to TSC1 or TSC2 mutations [1,2], leading to a cascade process of cell growth and metabolism [3]. However, the inhibition of the mTORC1-signaling pathway with a selective drug, rapamycin, demonstrated limited therapeutic effects including therapeutic resistance and immunosuppression [4,5]. Thus, it is extremely urgent to find a non-mTORC1-dependent drug therapy for TSC. TSC2 mutations are more pathogenic and cause more severe disease than TSC1 mutations [6,7]. In our study, the disease model with TSC2 deletion was selected as the research object.

Cluster of differentiation 24 (CD24), a heavily glycosylated mucin-type glycosylphosphatidylinositol-anchored cell surface molecule, has been verified as highly presented and associated with poor outcomes in human cancers [8]. It has been reported that CD24 promotes immune evasion via coupling with sialic acid-binding lg-like lectin 10 (Siglec-10), a inhibitory receptor expressed on tumor-associated macrophages [9]. CD24, like CD47, has been identified as an essential immunological checkpoint that also regulates several signaling pathways. In gastric cancer cells, CD24 positively regulates the expression of EGFR, supporting the EGFR-PI3K/Akt and EGFR-ERK signaling pathway to suppress EGFR internalization and degradation in a Rho-A related signaling axis [10].

The human epidermal growth factor receptor (EGFR or HER1) is a member of the tyrosine kinase family, and the homodimerization and heterodimerization of HER proteins activate downstream signaling cascades that promote cell survival and division while inhibiting apoptosis [11,12]. RUNX1/EGFR/STAT3 signaling pathway should be considered as a target for the therapy of TSC, because the abnormal activation of mTORC1 enhances the signal of RUNX1/EGFR/STAT3 to promote tumorigenesis [13]. Pyrotinib is an irreversible dual pan-ErbB receptor tyrosine kinase inhibitor, and it is the drug that acquired the first global approval for the therapy of breast cancer with HER2-positive expression [14]. Similarly, in non-small cell lung cancer (NSCLC) patients with high expression of HER2, pyrotinib offered a manageable safety profile with efficient antitumor efficacy [15]. However, it is unknown if pyrotinib has a particular effect on TSC patients with high EGFR expression.

The specific mechanisms by which pyrotinib targets TSC2-deficient tumor cells were investigated in this study, as well as the interaction between EGFR and CD24.

## 2. Materials and Methods

### 2.1. Cell Culture and Treatment

Mouse embryonic fibroblasts (MEFs) derived TSC2^−/−^ and TSC2^+/+^ cells donated by Professor Zha Xiaojun of Anhui Medical University, LAM patient-derived 621-101 and 621-103 cells and Eker rat uterine leiomyoma-derived (ELT3) cells acquired from Professor Xu Kaifeng of Peking Union Medical College Hospital, were cultured in DMEM medium, containing 10% fetal bovine serum (FBS) and 1% pen-strep (Solarbio, P7630, Peking, China). The macrophages of RAW264.7 and THP1 were cultured in 1640 medium. The cells were used prior to passage 20, and maintained in the incubator with stable temperature of 37 °C and humidified atmosphere comprising 5% CO_2_.

Lipo2000 (Invitrogen, 11668027, Waltham, MA, USA) was used as reagent of lipofection to perform siRNA interference technology based on the protocol of manufacturer. Small RNA of siCD24 and siControl were purchased from Guangzhou RiboBioCo., LTD (China), the sequence of CCACGCAGATTTACTGCAA was designed as siCD24 to hinder gene expression. Pyrotinib (HENGRUI, Jiangsu, China) dissolved to 50 mM in DMSO, and incubated cells with 1 or 5 μM. The inhibitor of STAT1 (MedChemExpress, HY-B0069, Monmouth Junction, NJ, USA) and STAT3 (MedChemExpress, HY-N0174, Monmouth Junction, NJ, USA) was added to cells with 0.5 or 1 μM. Gefitinib (MedChemExpress, HY-50895, Monmouth Junction, NJ, USA) was added with 50 μM. All inhibitors were dissolved to 50 mM in DMSO. Cells were treated with drugs in medium containing 1% FBS, and harvested after 48 h, except for cell cycle assay, which was performed in medium containing 10% FBS.

### 2.2. Cell Viability Assay

Approximately 4000 cells per well were seeded in a 96-well plate and treated with or without drugs after 24 h. Cell proliferation was monitored using the Cell Counting Kit-8 (CCK-8; MeilunBio, MA0218, Suzhou, China). Propidium iodide (PI; Solarbio, C0080, Peking, China, 5 μmol/L) was used to detect cell death. The data is presented as the percentage of dead cells relative to the total number of cells as detected by crystal violet staining. Values are expressed as the means ± SEM; *n* = 6/group.

### 2.3. Wound-Healing Assays

TSC2^−/−^ MEF cells were plated with about 1 × 10^5^ cells per well in a 6-well plate. In the center of wells, a 200 μL micropipette tip was used to make a vertical scratch. After cells were cultured with or without pyrotinib (1 μM) for 0, 8 and 24 h, the scratch width in each well was observed and the pictures corresponding to different time periods obtained by light microscope. The data was analyzed with Image-J software.

### 2.4. Immunoblotting Analysis

Total protein was obtained from processed cells in 6-well plates or tumor tissue, 15 μg protein was loaded into individual wells of 12% SDS-PAGE to resolve. Membranes were blocked in 3% bovine serum albumin (BSA) after electroblotting onto nitrocellulose membranes (Millipore, HATF00010, Billerica, MA, USA), then incubated overnight with antibodies against Tuberin/TSC2 (1:1000, CST, #4308, Danvers, MA, USA), EGFR (1:1000, Santa Cruz Biotechnology, sc-03, Dallas, TX, USA), pS6 (1:1000, CST, #4858, USA), α-tubulin (1:1000, Neomarker, ms581-P0, Fremont, CA, USA), pEGFR (1:1000, Santa Cruz Biotechnology, sc-57545, USA), HER2 (1:1000, Santa Cruz Biotechnology, sc-33684, USA), CDK4 (1:1000, Santa Cruz Biotechnology, sc-23896, USA), CDK2 (1:1000, Santa Cruz Biotechnology, sc-6248, USA), CD24 (1:1000, Santa Cruz Biotechnology, sc-19585, USA), STAT1 (1:1000, Santa Cruz Biotechnology, sc-464, USA), STAT3 (1:1000, Santa Cruz Biotechnology, sc-8019, USA), pSTAT1 (1:1000, Santa Cruz Biotechnology, sc-51700, USA), pSTAT3 (1:1000, Santa Cruz Biotechnology, sc-8059, USA), β-actin (1:1000, Santa Cruz Biotechnology, sc-47778, USA), cl-PARP (1:1000, CST, #9544S, USA), bcl2 (1:1000, Santa Cruz Biotechnology, sc-7382, USA), Siglec-10 (1:1000, Bioss, bs-2706R, Peking, China) and Lamin B1 (1:1000, Santa Cruz Biotechnology, sc-377000, USA) at 4 °C. After washing three times, the membranes were probed at room temperature using peroxidase-conjugated secondary antibodies (1:5000, Proteintech, B900620 & B900610, Chicago, IL, USA) for 1 h. The TANON image software (Beijing YuanPingHao Biotech, Peking, China) was used to detect the enhanced chemiluminescence (Millipore, WBKlS0100, USA) of specific membranes. Phosphorylated proteins were normalized to total proteins, nuclear and plasma proteins normalized to Lamin B1 and α-tubulin, respectively, and other proteins normalized to β-actin. All protein content was expressed in relative units in comparison with control samples loaded on each gel.

### 2.5. Real-Time qPCR and Primers

Total mRNA was extracted with TRIzol reagent (CWBIO, CW0580S, Peking, China) and reversed with EasyScript First-Strand cDNA Synthesis SuperMix (TransGen, AE301-02, Peking, China). UltraSYBR Mixture (CWBIO, CW0957C, Peking, China) was used to perform real-time qPCR with 50 ng cDNA per well. Melting curve analysis was used to confirm the specificity of primers. Difference of gene expression was analyzed with 2^−ΔCt^ method. The primer sequences of genes in this study were followed: CD24, 5′-GTTGCACCGTTTCCCGGTAA-3′(F), 5′-CCCCTCTGGTGGTAGCGTTA-3′(R); CDK2, 5′-CCTGCTTATCAATGCAGAGGG-3′(F), 5′-TGCGGGTCACCATTTCAGC-3′(R); β-actin, 5′-GGCTGTATTCCCCTCCATCG-3′(F), 5′-CCAGTTGGTAACAATGCCATGT-3′(R).

### 2.6. Confocal Microscopy

Cells were treated after seeding about 5000 cells on slides. Briefly, slides were fixed in 75% alcohol for 30 min at room temperature, rinsed with PBS and blocked with 1% BSA diluted in PBS for 1 h at room temperature. Then, the primary antibodies of pEGFR (1:100) were incubated with slides overnight at 4 °C. Slides were incubated with secondary antibodies conjugated with FITC for 1 h at room temperature after washing with PBS. DAPI (Solarbio, S2110, Peking, China, 10 μg/mL) was used to stain nucleus. Then, slides were washed and mounted with anti-fade mounting medium, and the fluorescence intensity of DAPI and FITC was revealed by confocal fluorescence microscope (Leica, TCS SP8, Allendale, NJ, USA).

TUNEL assay was performed with tumor tissue sections according to the manufacturer’s protocol (Yeasen, 40306ES20, Shanghai, China) and revealed by confocal fluorescence microscope (Leica, TCS SP8, Allendale, NJ, USA).

The representative images by 630× magnification were presented.

### 2.7. Chromatin Immunoprecipitation (ChIP) and Co-IP Assay

ChIP Chromatin Immunoprecipitation Kit (absin, abs50034, Shanghai, China) was used to perform ChIP assay, complexes of DNA and protein antibody were pulled down by pSTAT3 antibodies. Primers for the region of STAT3 binding motif (nucleotides −1944 to −1553) in CD24 promoter sequences was designed as 5′-CCTGCCATTACAGTCTTACC -3′ and 5′- CTGCTCCTCCAGCTTTCC-3′, and the size of production was expected to 392 bp.

Measures of 1 μg of anti-EGFR and anti-pEGFR antibodies were incubated with cell lysate after different treatments, and captured with protein-A/G agarose (Santa Cruz Biotechnology, sc-2003, Dallas, TX, USA) overnight at 4 °C. The beads were washed 5 times and suspended with SDS loading buffer, then centrifuged after boiling to acquire the supernatant for immunoblotting analysis.

### 2.8. Cytosolic and Nuclear Extractions

Extracts of cytosolic and nuclear were acquired by nuclear and cytoplasmic extraction kit (Solarbio, SN0020, Peking, China), following the instructions for use.

### 2.9. Immunohistochemistry Staining

The tumor tissues were taken from sacrificed nude mice, tissue sections were obtained after successively fixing with 4% formaldehyde, dehydrated with gradient (70%, 80%, 95% and 100%) alcohol and xylene, and embedded in paraffin. Immunohistochemical staining were performed with the paraffin sections according to DAB working solution (Solarbio, DA1010, Peking, China), and hematoxylin was used to stain cell nucleus. Slides were incubated with anti-pEGFR, anti-CD24, anti-CDK2, anti-CDK4 and anti-ki67 (CST, #9449, Danvers, MA, USA) antibodies, which were diluted at 1:100, and imaged at 400× magnification.

### 2.10. Flow Cytometry Analysis

TSC2^−/−^ MEF cells treated with or without 1 µM pyrotinib were collected from 6-well plates. Cells were stained with primary anti-CD24 antibodies (1:200) for 2 h at 4 °C, and then incubated FITC-conjugated secondary antibodies for 1 h at room temperature. Finally, flow cytometry analysis of cells was performed on a BD FACS Aria II flow cytometer (BD Biosciences, LSR Fortessa, Franklin, NJ, USA).

Apoptosis (MULTI SCIENCES, AP105, Hangzhou, China) and cell cycle (MULTI SCIENCES, CCS012, Hangzhou, China) analysis were performed with kits based on the manufacturer’s instructions.

### 2.11. Animal Experiments

The ethics committee for animal use at the medical college of Nankai University approved in vivo experiments. Equal amounts of TSC2^+/+^ and TSC2^−/−^ MEF cells (2 × 10^6^) were subcutaneously injected into both flanks of Balb/c nude mice (5 per group), and tumor tissues were collected to perform IHC after 1 month.

Tumor cells TSC2^−/−^ MEF (2 × 10^6^) were subcutaneously injected into both flanks of Balb/c nude mice, and the mice were randomized into 2 groups (5 mice per group) after ten days. The administration group was treated with pyrotinib (40 mg/kg, *i.g.*), diluted in normal saline. The control group was treated with an equal volume of normal saline; the medicine was given twice a week. Primary tumors were measured using Vernier calipers. Mice were given the substrate D-luciferin (PerkinElmer, #122799, Waltham, MA, USA) by intraperitoneal injection (150 mg/kg), and tumors were assessed using bioluminescence imaging (BLI; PerkinElmer, IVIS Spectrum, USA) every week. Animals were head-fixed and anaesthetized with isoflurane (~1%) during imaging. The formula, volume  =  Length × Width^2^/2, was used to monitor and calculate the tumor volume. After 4 weeks, tumors were harvested for the following experiments.

### 2.12. Statistical Analysis

All data are showed as the mean  ±  SEM of at least 3 independent experiments and were analyzed using GraphPad Prism 8.0 statistical software. Student’s t-test was used to compare significance of the difference between two groups, and two-way ANOVA was used for multiple-group comparisons. *p* < 0.05 was considered significant.

## 3. Results

### 3.1. Pyrotinib Specifically Reduced Cellular Viability of TSC2-Deficient Cells

Tuberous sclerosis complex is characterized by a mutation in TSC1 or TSC2 which activates the signal pathway of mTORC1/RUNX1/EGFR with loss of function [2,13]. In a phase I trial, pyrotinib, as an irreversible inhibitor of EGFR, exerted viable antitumor efficiency and acceptable tolerability [16]. We performed the mouse embryonic fibroblast (MEF) cell model experiment to examine the expression of tyrosine kinase receptors. The results showed that TSC2-deficient cells had high levels of HER2 and EGFR expression, as well as aberrant activation of pS6, a marker of mTORC1 activation (Figure 1A). To determine whether high EGFR expression enabled pyrotinib to be targeted selectively, different concentrations of pyrotinib were used. Pyrotinib was highly selective towards TSC2-deficient MEF cells, which demonstrated poorer cell viability (Figure 1B) and greater levels of cell death (Appendix A) compared with control cells. In the study of pyrotinib selectively targeting TSC2-null cells, compared with LAM and ELT3 cells, TSC2-deficient MEF cells were more sensitive to pyrotinib without affecting the viability of TSC2-added cells. Therefore, MEFs was identified as the main disease model in our study. The high expression of pEGFR/EGFR, the cyclin-dependent kinase CDK2 and CDK4 in TSC2-deficient cells was effectively suppressed by pyrotinib (Figure 1C). Interestingly, the level of pS6 did not change, suggesting that pyrotinib targeted TSC2-deficient cells in mTORC1-independent manner, inconsistent with the effect of rapamycin [17]. Pyrotinib induced G1 phase arrest (Figure 1D), and hindered cell migration in TSC2-deficient cells (Figure 1E), which was consistent with the protein data. These findings verified that pyrotinib slowed cell proliferation and migration, and raised the question of whether pyrotinib killed TSC2-deficient cells directly. Further studies showed that pyrotinib induced early and late apoptosis in TSC2-deficient cells by dual staining with 7AAD and Annexin V-APC (Figure 1F). Pyrotinib inhibited cell viability and triggered apoptosis in TSC2-deficient cells, and demonstrated a high level of targeted destruction in both human and rat TSC2-deficient cell models, bolstering the validity of MEF model results (Appendix A).

### 3.2. Elevated CD24 Level Promoted Cell Vitality of TSC2-Deficient Cells

Given that drug-induced immunosuppression is one of the most challenging aspects of TSC therapy, immune-related gene expression was examined in TSC2 or TSC2-null MEF cells [13]. The differential CD24 expression stood out among numerous genes in the RNA-seq results (Appendix A). To verify this result, the MEF cells with or without TSC2 were subcutaneously implanted in nude mice. The IHC results of tumor tissue indicated that pEGFR and pS6 were highly expressed in TSC2-depleted tumor tissue, and CD24 exhibited considerable positive expression (Figure 2A). Whether LAM-derived or MEF-derived cell lines were used in the protein tests, the results were comparable (Figure 2B). CD24 facilitated immune evasion by interacting with the inhibitory receptor siglec-10 of tumor-associated macrophages [9]. Siglec-10 expression of macrophage was markedly positive in the TSC2-deficient group after treatment with different culture supernatants (Figure 2C). Therefore, the microenvironment of TSC2-deficient tumor cell might promote high expression of CD24 in macrophages to induce autoimmune escape. Given that EGFR and CD24 were highly expressed simultaneously, pyrotinib was used to explore the correlation between the two. Following pyrotinib treatment, CD24 was inhibited both at gene and protein levels (Figure 2D,E), and cell cycle-related gene and proteins were restrained in the TSC2-deficient cells with CD24 silenced (Figure 2F,G); the data was consistent with pyrotinib therapy. Furthermore, pyrotinib treatment or CD24 silencing both resulted in the decreased expression of siglec-10 in macrophages co-incubated with the cultured tumor cell supernatant (Figure 2H,I). In the LAM-derived tumor model, pyrotinib was also shown to inhibit the expression of CD24 and siglec-10 in the tumor microenvironment (Appendix A). Pyrotinib restrained the cell cycle and reduced immunosuppressive responses partly by inhibiting CD24. Apoptosis was promoted by either pyrotinib or CD24 knockdown, as evidenced by a decrease in the anti-apoptotic protein bcl2 (BCL2 apoptosis regulator) and an increase in the pro-apoptotic protein cl-PARP (Figure 2J,K). From this, we concluded that pyrotinib targeted TSC2-deficient cells to induce growth inhibition and apoptosis might via CD24.

### 3.3. CD24 Transcriptional Level Upregulated by Phosphor-STAT3

The phosphorylation of the EGFR intracellular kinase domain was reduced by pyrotinib. Changes of pEGFR downstream STAT-related proteins and their phosphorylation levels [18] were detected to explore how pyrotinib regulates CD24. In both LAM-derived and MEF-derived TSC2-deficient cells, Western blotting indicated higher baseline production of STAT1/STAT3 total protein and phosphorylated protein, which could be restrained by pyrotinib (Figure 3A). This led to the hypothesis that pyrotinib inhibits CD24 by inhibiting pEGFR/pSTAT signaling. TSC2-depleted tumor cells were shown to be susceptible to both the pSTAT1 inhibitor (fludarabine) [19] and the pSTAT3 inhibitor (cryptotanshinone) [20] (Figure 3B). However, only cryptotanshinone inhibited the gene expression of CD24 as pyrotinib did, and fludarabine made no reference to the expression of CD24 (Figure 3C). Similarly, cryptotanshinone induced significant early and late apoptosis in TSC2-deficient cells, but the inhibitor of pSTAT1 was not associated with apoptosis (Figure 3D), which led us to wonder whether the inhibition of CD24 by pyrotinib is achieved through the pEGFR/pSTAT3 signaling axis. To this end, we queried the promoter sequence of CD24 in the ensemble online website (https://asia.ensembl.org/index.html, accessed on 16 January 2022) to detect whether STAT3, as a general transcription factor, could directly transcribed and promoted the expression of CD24. Potential binding sites for STAT3 (TTCCATGAAA) [21] were discovered in the promoter region of CD24 (Appendix A). The former and revised primers were designed around the potential binding motif, and the results of ChIP PCR illustrated that pSTAT3 could bind to the promoter region of CD24 (Figure 3E). Pyrotinib could suppress the pEGFR/pSTAT3 signaling pathway, and the inhibition of pSTAT3 further reduced the transcriptional expression of CD24. Given that pyrotinib was a pan-inhibitor of EGFR/HER2, another EGFR inhibitor, gefitinib was used to avoid an off-target effect. The results showed that pSTAT3/CD24 was significantly restricted with the phosphorylation of EGFR inhibited, accompanied by decreased proliferation and increased apoptosis (Appendix A).

### 3.4. Binding with CD24 Enhanced Phosphor-EGFR Function

Phosphorylation of EGFR promotes cell growth by activating downstream signaling pathways [22]. However, pyrotinib covalently binds to the intracellular kinase region of the HER family via its ATP binding sites, which prevents the formation of HER family dimer, inhibits phosphorylation itself and the downstream signaling pathway [23]. The above results indicated that pyrotinib inhibited phosphorylation of EGFR, and subsequent confocal analysis revealed that pyrotinib hindered both protein expression and nuclear transport of pEGFR (Figure 4A and Appendix A).

To explore how pEGFR regulates pSTAT3 in cells, we separated the cytoplasm and nucleus of the cells. α-tubulin and Lamin B1 were used as markers of the cytoplasm and nucleus, respectively. As expected, the nucleus import of pSTAT3 was inhibited along with pEGFR after pyrotinib treatment. Interestingly, we found that CD24 was simultaneously presented in the nucleus and cytoplasm, and concomitant with the reduction of pEGFR in the nucleus, CD24 expression was inhibited by pyrotinib in both cytoplasm and nucleus (Figure 4B). Therefore, is there a direct contact between pEGFR and CD24? Co-IP tests were carried out on two proteins. The results elucidated that pyrotinib not only inhibited the expression of the total CD24 protein in TSC2-deficient cells, but also drastically reduced CD24 pulling down by pEGFR in the same quantity of protein (Figure 4C). Meanwhile, co-IP experiment after nucleocytoplasmic separation showed that CD24 accumulated in the cytoplasm after pyrotinib inhibited the entry of pEGFR into the nucleus, but the binding between CD24 and EGFR was not significant (Figure 4D). In order to further confirm whether the accumulation of CD24 was in the cell membrane or in the free cytoplasm, the expression of CD24 on the cell membrane was measured by flow cytometry, and the mean fluorescence intensity of CD24 was reduced following pyrotinib treatment (Figure 4E). Therefore, CD24 not only played an immune escape function on the cell membrane, but also existed in the cytoplasm and nucleus, and could directly bind to pEGFR. In order to explore whether the combination of CD24 and pEGFR in the cytoplasm and nucleus is meaningful, the expression of CD24 was suppressed in TSC2-deficient cells. Conversely, knockdown of CD24 restrained the phosphorylation levels of STAT1 and STAT3, which are pEGFR downstream proteins (Figure 4F). Thus, the direct binding between pEGFR and CD24 could affect the function of pEGFR. In general, EGFR and pEGFR were abnormally expressed in TSC2-deficient cells, thereby activating pSTAT3 to directly transcribe and increase the expression of CD24, in turn, CD24 could also bind to pEGFR to assist its function, establishing a positive feedback loop that could be targeted by pyrotinib (Figure 4G).

### 3.5. Pyrotinib Impeded the Viability of TSC2 Deficient Cells In Vivo

It is a pity that human samples were difficult to acquire because of the rarity of TSC cases. To further confirm the efficiency of pyrotinib in vitro, Balb/c nude mice were selected as experimental animals to construct the mouse xenograft model. Mice were randomly divided into pyrotinib-administered and control groups after subcutaneous tumor-bearing using TSC2-deficient MEF cells. The administration group was given pyrotinib (40 mg/kg/day) by gavage, while the control group was treated with normal saline. The average tumor fluorescence intensity was assessed by bioluminescence imaging (BLI) every week. The results revealed that the fluorescence intensity of the pyrotinib-administered group gradually declined relative to the control group, and there was an obvious difference in the fourth week (Figure 5A). Consistent with this, the tumor volume and weight of pyrotinib-administered group were considerably lower than the control group (Figure 5B,C), and corresponding to this result, the TUNEL assay showed that pyrotinib induced apoptosis of tumor tissue cells (Figure 5D). Pyrotinib inhibited tumor cell viability and induced apoptosis in vitro and in vivo. Immunohistochemical results indicated that Pyrotinib reduced the expression of ki67, CDK2, CDK4 and CD24 (Figure 5E and Appendix A). According to the immunoblotting of tumor tissue, pyrotinib suppressed the pEGFR/pSTAT3 signaling pathway, decreased CD24 protein expression, and produced alterations in apoptosis-related proteins cl-PARP and bcl2, which fully echoed the mechanism of the above in vitro experiments (Figure 5F). Together, pyrotinib inhibited tumor development by inducing apoptosis and suppressing CD24 expression via the pEGFR/pSTAT3 signaling pathway.

## 4. Discussion

Tuberous sclerosis complex, an autosomal dominant disorder, characterized by mutations in TSC1 or TSC2, causes hamartoma formation in various of the organ systems [24]. Considering TSC mutation leading to the downstream mTOR signaling pathway maladjustment, the application of mTOR inhibitor, everolimus, has exerted outstanding therapeutic benefits in TSC-related progression of the disease [25]. Rapalogs including rapamycin and everolimus were originally designed to treat cancer and organ transplant rejection, but with poor pro-apoptotic activity and cell cytostatic capacity [26,27]. Therefore, it is urgent to investigate other therapies for TSC-related disease in a rapalog-independent manner.

Research has shown that TSC2^−/−^ ASM cell proliferation is EGF-dependent [28]. Anti-EGFR treatment restrained lymphangiogenesis [29], and the reversed pulmonary alterations had no side effects with lung tissue, making it more specific and effective than rapamycin [30]. Targeting tyrosine kinase has shown promise in the treatment of malignant disorders [31]. Consistent with these studies, pyrotinib targeted TSC2-deficient cells with high EGFR expression, reduced cell proliferation and induced apoptosis by inhibiting EGFR phosphorylation. The growth inhibition of HER2 antagonists is achieved by restraining PI3K and mTOR signals [32]. However, whether the combination of pyrotinib and rapamycin would more effective in clinical treatment needs further study.

In view of the immunosuppressive effect of rapamycin in clinical treatment, we focused on the expression changes of immune-related molecules in TSC2-deficient cells and found a significantly high expression of CD24. Surprisingly, pyrotinib not only inhibited CD24 expression, but also reduced cyclin proteins and accumulated apoptotic proteins, achieving the same effect as inhibiting EGFR phosphorylation. In breast cancer, higher CD24 expression is significantly associated with a lower overall and disease-free survival rate, which suggests that CD24 is an efficient prognostic factor, closely related to clinicopathological factors, including lymph node infiltration and TNM staging [33]. It points to the ability of CD24 to regulate RhoA/EGFR signaling [10], and to increase the expression of HER2 by activating NF-κB transcription factor [34]. G7mAb, an anti-CD24 antibody derived by hybridoma technology, was reported to attenuate phosphorylation of Src/STAT3, thus enhancing the antitumor efficiency of cetuximab [35]. All these data indicate that CD24 is the upstream regulator of the expression of tyrosine kinase receptors such as EGFR and thus affects cell viability. Interestingly, our study seems to be a paradox, so what is the regulatory network between CD24 and EGFR? After separating the nucleus and cytoplasm of cells, it was surprising that CD24 was not only expressed in the cytoplasm or membrane, but also detected in the nucleus, and its expression trend was consistent with pEGFR, thus, we found that pEGFR and CD24 could be directly combined.

Insulin-like growth factor binding protein 2 (IGFBP2), which has both extracellular and intracellular functions, increases nuclear accumulation of EGFR, thereby enhancing the transcriptional activation of STAT3 [36]. CD24, as a membrane protein, is involved in immune escape [9] and cancer cell stemness [37], but could CD24 in the cytoplasm and nucleus play a biological function? On the other hand, six-transmembrane epithelial antigen of the prostate 3 (STEAP3) is involved in regulating STAT3 transcriptional activation mediated by EGFR in a manner of positive feedback, accompanied by upregulating the expression and nuclear trafficking of EGFR [38]. Existing research showed that pyrotinib suppressed HER2 nuclear transport [39], and CD24 was increased by stable overexpression of HER2 [40]. This is consistent with our findings that pyrotinib restrained the nuclear translocation of pEGFR and simultaneously reduced the nucleocytoplasmic ratio of CD24. The interaction between the two molecules further suppressed the activation level of STAT3, and in turn the downregulated STAT3 directly inhibits the transcription CD24 expression, thus forming a positive feedback pathway amongst these molecules. However, the specific mechanism of how CD24 assists pEGFR still needs further investigation.

## Figures and Tables

**Figure 1 cells-11-03064-f001:**
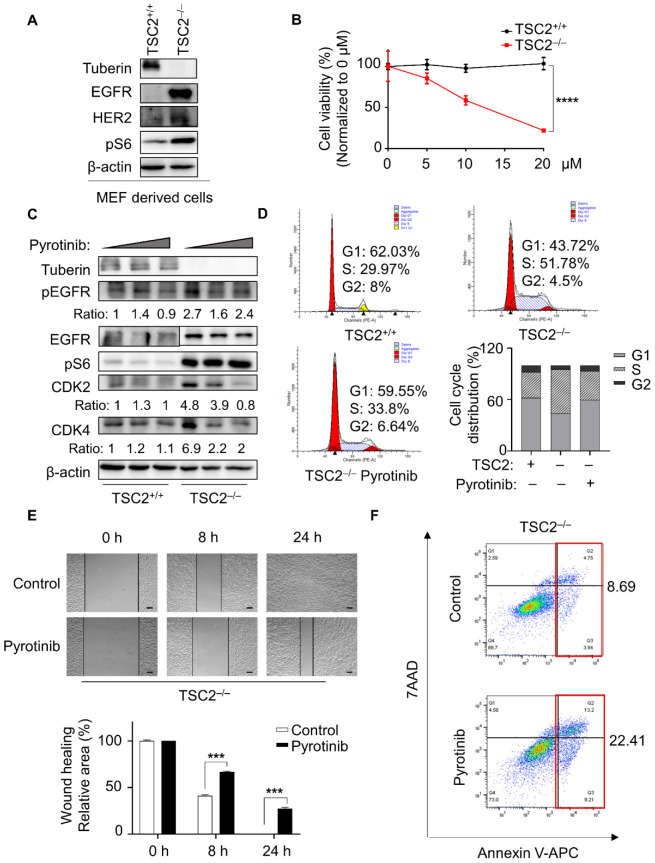
Pyrotinib specifically reduced cellular viability of TSC2-deficient cells. (**A**) MEF-derived cells with or without TSC2 expression were lysed, and indicated proteins were detected by immunoblot. (**B**) MEF-derived cells were dealt with different concentrations of pyrotinib for 48 h, and cell viability was monitored using the CCK8 assay (*n* = 6), the half maximal inhibitory concentration (IC50) was calculated after pyrotinib treatment in TSC2-deficient cells (IC50 = 11.5). **** *p* < 0.0001, TSC2^+/+^ versus TSC2^−/−^ at 20 µM, Student *t* test. (**C**) Immunoblot analysis of MEF-derived cells treated or not treated with pyrotinib (1 µM or 5 µM). (**D**) Cell cycle distribution was detected in MEF-derived cells treated with or without pyrotinib (1 µM), fluorescence signals were collected by flow cytometry. (**E**) TSC2-deficient cells grown to confluent monolayer were scratched and treated with pyrotinib (1 µM) for 8 h and 24 h. Representative images at the 0, 8 and 24 h time points are shown (magnification 40×, bar = 200 µm) *** *p* < 0.001, control vs. pyrotinib at 8 h or 24 h, Student *t* test. (**F**) TSC2-deficient MEF-derived cells were treated with pyrotinib (1 µM) for 48 h. Annexin V: FITC Apoptosis Detection Kit (MULTI SCIENCES) was used to stain cells. Anoikis was determined by flow cytometry. Annexin V^+^/7AAD^+^ indicated cells undergoing late apoptosis and Annexin V^+^/7AAD^−^ indicated early apoptosis.

**Figure 2 cells-11-03064-f002:**
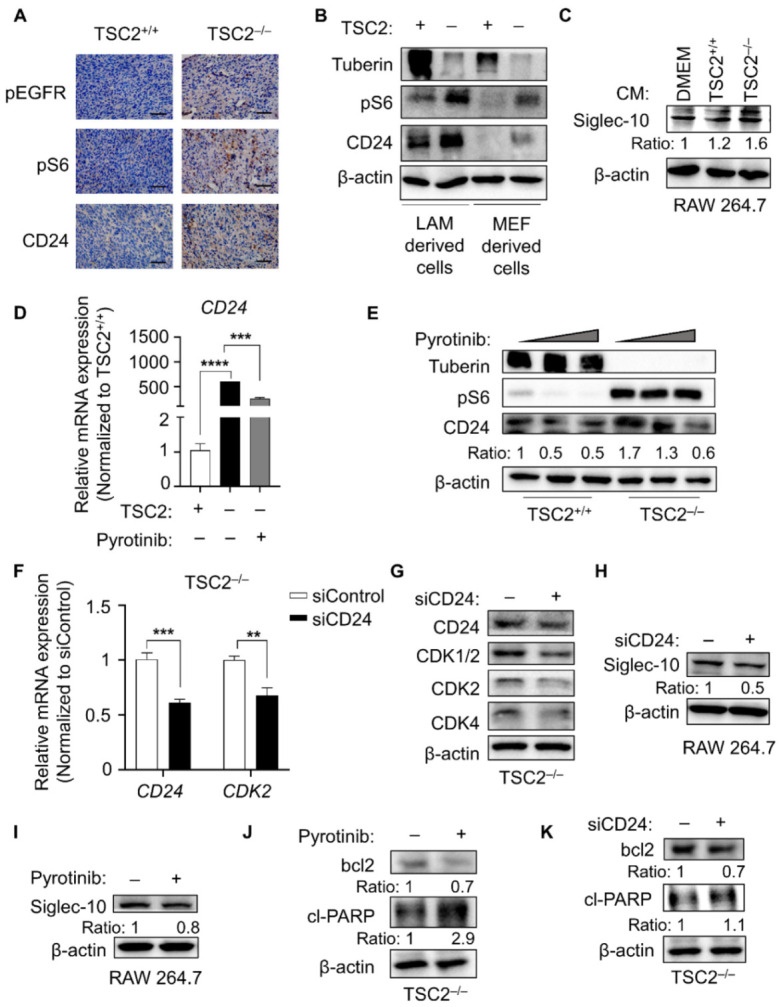
Elevated CD24 level promoted cell vitality of TSC2-deficient cells. (**A**) Immunohistochemical staining of pEGFR, pS6 and CD24 in tumor tissues, which were taken from nude mice following tumor bearing with MEF-derived cells (magnification 400×, bar = 50 µm). (**B**) Immunoblot analysis of proteins in LAM or MEF-derived cells. (**C**) Macrophages of RAW 264.7 were co-incubated with DMEM and culture supernatant of MEF with or without TSC2-deficient cells for 6 h, and RAW 264.7 cells were lysed, and siglec-10 was detected by immunoblot. (**D**) *CD24* mRNA expression level was detected by qPCR (*n* = 3) in MEF-derived cells treated with or without pyrotinib (1 µM). **** *p* < 0.0001, TSC2^+/+^ vs. TSC2^−/−^. *** *p* < 0.001, TSC2^−/−^ vs. TSC2^−/−^ + pyrotinib, Student *t* test. (**E**) Immunoblot analysis of MEF-derived cells treated with or without pyrotinib (1 µM or 5 µM). (**F**) *CD24* and *CDK2* mRNA expression level was detected by qPCR (*n* = 3) in TSC2-deficient cells transfected with siControl or siCD24. *** *p* < 0.001, siControl vs. siCD24 in the mRNA of *CD24*. ** *p* < 0.01, siControl vs. siCD24 in the mRNA of *CDK2*, Student *t* test. (**G**) Immunoblot analysis of proteins in TSC2-deficient cells transfected with siCD24 or not. (**H**,**I**) Macrophages of RAW 264.7 were co-incubated with culture supernatant of TSC2-deficient cells treated with or without siCD24 (**H**) or 1 µM pyrotinib (**I**), and siglec-10 was detected by immunoblot. (**J**,**K**) Apoptosis-related protein was detected by immunoblot in TSC2-deficient cells treated with or without 1 µM pyrotinib (**J**) or si*CD24* (**K**).

**Figure 3 cells-11-03064-f003:**
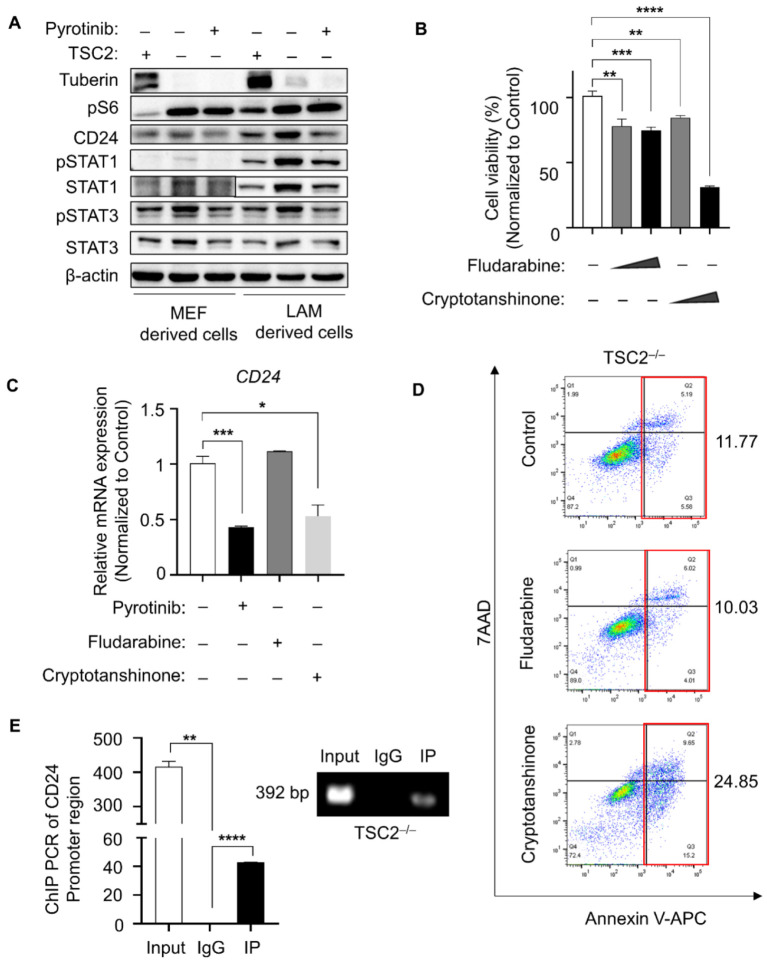
CD24 transcriptional level upregulated by phosphor-STAT3. (**A**) Indicated proteins were detected in MEF- and LAM-derived cells treated with or without pyrotinib (1 µM). (**B**) TSC2-deficient MEF-derived cells were treated with fludarabine (0.5 µM or 1 µM) or cryptotanshinone (0.5 µM or 1 µM) for 48 h, CCK8 assay was used to monitor cell viability (*n* = 5). ** *p* < 0.01, control versus fludarabine (0.5 µM). *** *p* < 0.001, control vs. fludarabine (1 µM). ** *p* < 0.01, control vs. cryptotanshinone (0.5 µM). **** *p* < 0.0001, control vs. cryptotanshinone (1 µM), Student *t* test. (**C**) mRNA expression changes of *CD24* was detected by qPCR (*n* = 3) in TSC2-deficient cells treated with or without pyrotinib (1 µM), fludarabine (1 µM) and cryptotanshinone (1 µM). *** *p* < 0.001, Control vs. pyrotinib. * *p* < 0.05, control vs. cryptotanshinone, Student *t* test. (**D**) TSC2-deficient MEF-derived cells were treated with fludarabine (1 µM) or cryptotanshinone (1 µM) for 48 h. Cells were stained with the Kit. (**E**) RT-PCR was performed to determine gene abundance of CD24 promoter region in the groups of positive control (Input), negative control (IgG) and IP, which were immunoprecipitated with anti-pSTAT3 antibody in TSC2-deficient MEF-derived cells. ** *p* < 0.01, input vs. IgG. **** *p* < 0.0001, IgG vs. IP, Student *t* test.

**Figure 4 cells-11-03064-f004:**
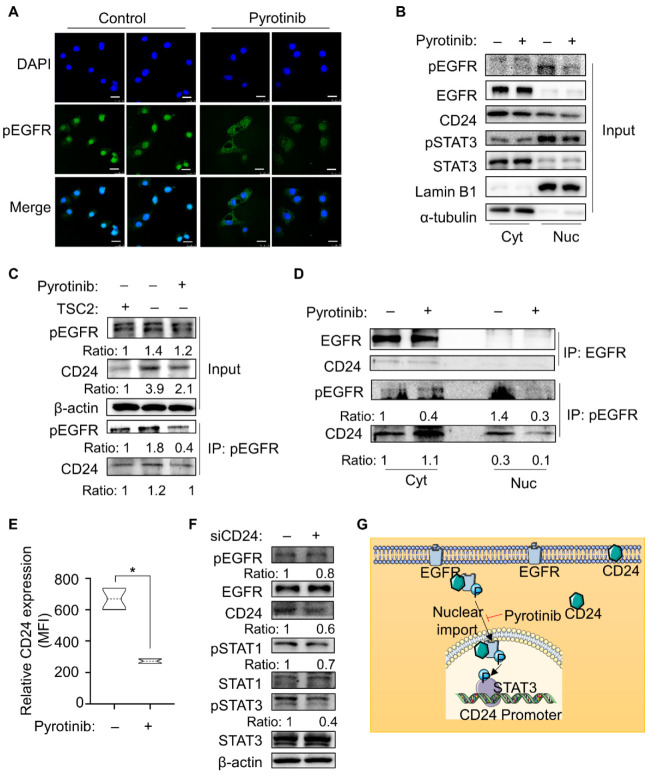
Binding with CD24 enhanced phosphor-EGFR function. (**A**) Localization and expression of pEGFR revealed by immunofluorescence staining in TSC2-deficient MEF-derived cells treated with or without pyrotinib (1 µM). pEGFR (FITC-labeled) was expressed on both the cytoplasm and the nuclear, treatment with pyrotinib diffused the nuclear-accumulated pEGFR into the cytoplasm. DAPI was used for nuclear staining (magnification 630×, bar = 25 µm). (**B**) Cells were treated with pyrotinib (1 µM) or not. Immunoblot analysis was used to detect indicated proteins (input) from the cytosolic and nuclear extracts. (**C**) MEF-derived cells treated with or without pyrotinib (1 µM), and immunoblot analysis was used to detect proteins of cell lysates after incubated with antibodies of pEGFR (IP) or not (input). (**D**) Cells were treated with pyrotinib (1 µM) or not. The cytosolic and nuclear extracts were subjected to immunoblot analysis using antibodies against EGFR or pEGFR (IP). (**E**) Flow cytometry was used to measure the mean fluorescence intensity of CD24 in TSC2-deficient cells treated with or without pyrotinib (1 µM), which was co-incubated with CD24 antibodies (*n* = 3). * *p* < 0.05, Student *t* test. (**F**) Immunoblot analysis of proteins in TSC2-deficient cells transfected with siCD24 or not. (**G**) Schematic illustration of this study. Pyrotinib restrained phosphorylation and nuclear translocation of pEGFR and directly suppressed the expression level of CD24 by inhibiting pEGFR/pSTAT3, pSTAT3 transcriptionally binding with the promoter region of CD24, and pEGFR/pSTAT3 pathway reduced the nucleocytoplasmic ratio of CD24, thus forming a positive feedback pathway.

**Figure 5 cells-11-03064-f005:**
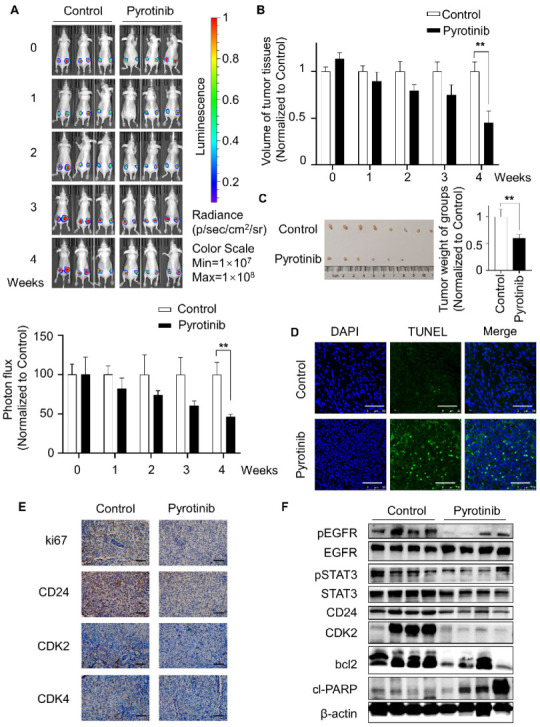
Pyrotinib impeded the viability of TSC2 deficient cells in vivo. (**A**) After subcutaneous injection of luciferase-expressing TSC2-deficient cells into female nude mice, pyrotinib (40 mg/kg) therapy or normal saline were administered through gavage after 10 days (*n* = 4). The bioluminescence images were taken at 0–4 weeks, bioluminescence intensity was recorded and quantified. ** *p* < 0.01, control vs. pyrotinib in the fourth week, Student *t* test. (**B**) Orthotopic xenograft tumor size was measured at 0–4 weeks. ** *p* < 0.01, control vs. pyrotinib in the fourth week, Student *t* test. (**C**) Representative tumor images from different groups of mice. ** *p* < 0.01, control vs. pyrotinib, Student *t* test. (**D**) TUNEL-positive cells were performed by kit, DAPI was used for nuclear staining (magnification 630×, bar = 50 µm). (**E**) Immunohistochemical staining of ki67, CD24, CDK2 and CDK4 in tumor tissues of different groups (magnification 400×, bar = 50 µm). (**F**) Indicated proteins were detected in tumor tissues from groups.

## Data Availability

All data generated or analysed during this study are included in this published article and are available from the corresponding author upon reasonable request.

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
