# Peer review of "Pyrotinib Targeted EGFR-STAT3/CD24 Loop-Mediated Cell Viability in TSC"

_cells, 2022, doi:10.3390/cells11193064_

Round 1

Reviewer 1 Report

Xiao Han and the colleagues provided a manuscript about effect of pyrotinib on EGFR-STAT3 pathway in TSC. The authors provide interesting evidence of the role of CD24 in the development of TSC2-deficient tumors. These findings might contribute to the improvement of therapy regime for TSC patients in the future.

Main points:

1.       Fludarabine is not a STAT1 inhibitor, but rather a general chemotherapeutic agent inhibiting all dividing cells, therefor it use as STAT1 inhibitor is wrong. It is not surprising that there was no effect on apoptosis observed.

2.       Pyrotinib is described in the paper as pan-ERRB inhibitor, whereas majority of the literature refer to it as a selective EFGR/HER2 inhibitor. Moreover, to access the role of pyrotinib versus potential off-target effects, other EGFR inhibitors available on the market should be used to validate the observed effect.

3.       IC50 values of cell viability should be measured and disclosed for comparison of the drug effects.

4.       Proper introduction to TSC disease vs protein is missing. It should be clear why TSC2 not TSC1 deficient MEF were used.

5.       Scale bars should be added to all microscopic pictures.

6.       Control used for in vitro and in vivo drug treatments should be disclosed – DMSO, PBS, etc. Also formulation of the drugs should be clear.

7.       Figure 5 – absolute tumor volume numbers and radiance should be plotted over time. It is quite concerning that tumors to not appear to grow over time and at the moment of treatment start (10 days) pyrotinib group showed already reduced tumors. If available, images of all 5 mice/group should be shown. If some mice were excluded it needs to be clearly stated and argued.  

8.       Figure 1C- quantification of the CDK2 and CDK4 seems to not match the western blots. Possible mistake?

9.       Fig1F – in the figure it is written 621-101 cell line, but the legend states it is LAM-derived cell line.

10.   Figure 5D – low resolution of the images should be improved.

11.   Figure2D, Figure3B,C,E, etc – student t-test in not suitable for multiple comparisons, 1-way anova with multiple comparison correction should be used.

12.   Figure legends should clearly state which statistical test was used and number of biological/technical replicates should be disclosed.

13.   Lines 227-228 – overstatement based on only one experiment.

Minor points:

1.       Citation for Cryptotanshinone should be added.

2.       Line 352 – obvious, not obviously.

3.       Pyrotinib writing small vs capital needs to be unified.

4.       Fig1A consistent labeling TSC22+/+ and TSC2-/- would make understanding easier.

5.       Bioluminescence imaging – should be stated which machine, if luciferin was injected, if yes which one and how much.

6.       FigS1A – how exactly cell death is defined – proportion of PI+-cells?

7.       Line 303 – “expressed” is a wrong term

8.       For apoptosis, cell cycle, nuclear and cytoplasm fractioning kits exact kits names and catalog numbers should be stated.

9.       Antibodies (and all other reagents) catalog numbers should be disclosed.

10.   Material and methods – it should be disclosed which alcohol was used.

11.   Catalog number, dilution missing for CDK2, CDK4, siglec-10, cl PARP antibody.

12.   Lines 108,109 “Phosphorylated proteins were quantified by their total proteins..” – meaning unclear.

13.   Line 366 - pyrotinib typo.

Author Response

Dear Reviewer,

Thank you for considering our revised manuscript “Pyrotinib Targeted EGFR-STAT3/CD24 Loop-mediated Cell Viability in TSC” for the Journal of cells.

We addressed all comments to you.  The attached text and figures labelled in the marked version. Thanks for your consideration of our work!

Main points:

Question 1: Fludarabine is not a STAT1 inhibitor, but rather a general chemotherapeutic agent inhibiting all dividing cells, therefor it use as STAT1 inhibitor is wrong. It is not surprising that there was no effect on apoptosis observed.

Response 1: Thanks for your comments. We agreed that Fludarabine indeed inhibited DNA synthesis, and acted on non-dividing (G0 phase) cells (PMID: 12138643). However, more researches confirmed that Fludarabine as one of STAT1 inhibitors, which was widely used in the study of multiple disease models, such as cancer (PMID: 34462423), steroid-induced avascular necrosis of the femoral head (PMID: 28601953), periodontitis (PMID: 31329047), and so on. In addition, study further supported that Fludarabine restrained the activation of STAT1 and the transcription of STAT1-dependent gene in normal resting or activated lymphocytes, in which it caused a specific depletion of STAT1 protein (and mRNA) but not other STATs (PMID: 10202937). Therefore, we considered and used Fludarabine as an inhibitor of STAT1 in our work. Together with the result that Fludarabine failed to inhibit gene expression of CD24, whereas Pyrotinib and STAT3 inhibitors reduced CD24 transcription and induced apoptosis, thus demonstrating the important role of STAT3 highlighted in model of Pyrotinib inhibition for TSC.  And thank you for your suggestion, we have added corresponding reference on Fludarabine as a STAT1 inhibitor in the manuscript.

Question 2: Pyrotinib is described in the paper as pan-ERRB inhibitor, whereas majority of the literature refer to it as a selective EFGR/HER2 inhibitor. Moreover, to access the role of pyrotinib versus potential off-target effects, other EGFR inhibitors available on the market should be used to validate the observed effect.

Response 2: Thanks for your valuable advice. Pyrotinib is an irreversible inhibitor targeting tyrosine kinase of HER2/EGFR. However, in TSC related disease model, the abnormal expression of EGFR in TSC has been widely studied compared with HER2. There was a significant correlation between p-EGFR Tyr-1068, p-EGFR Tyr-992 with hamartin, as well as the p-mTOR and p-EGFR Tyr-1173 in adenocarcinoma, abnormal hamartin expression may be EGFR signaling dependent but is independent on EGFR mutations (PMID: 24593867). Anti-EGFR antibody efficiently and specifically inhibited human TSC2-/- smooth muscle cell proliferation, and it was considered to be a possible treatment option for TSC and LAM (PMID: 18958173). However, the correlation between HER2 and TSC has not been reported, thus we considered that the pharmacological effect of Pyrotinib on cells was mainly targeting EGFR. To further access the role of Pyrotinib versus EGFR, Gefitinib, another EGFR inhibitor was used. The results showed that Gefitinib significantly inhibited EGFR/pSTAT3/CD24 axis, thus attenuating cell proliferation and inducing apoptosis in TSC2-deficient cells. And the result was attached in Supplementary data (Fig. S4).

Question 3: IC50 values of cell viability should be measured and disclosed for comparison of the drug effects.

Response 3: Thank you for your question. According to your advice, we have measured the IC50 values of cell viability in TSC2-deficient cells. But the TSC2-expressing MEF cells were insensitive to Pyrotinib due to the low EGFR expression level, and the IC50 could not calculated (Fig. 1B). IC50 of LAM and ELT3 derived TSC2-deficient cells also calculated (Fig. S1B and S1C).

Question 4: Proper introduction to TSC disease vs protein is missing. It should be clear why TSC2 not TSC1 deficient MEF were used.

Response 4: Thanks for your valuable advice. In TSC disease, sporadic patients with TSC1 mutations had, on average, milder disease in comparison with patients with TSC2 mutations, despite being of similar age (PMID: 11112665). Through constructing the TSC1 and TSC2 variants, 16 of 45 TSC1 variants (36%) and 70 of 107 TSC2 variants (65%) classified as pathogenic, 29 of 45 TSC1 variants (64%) and 37 of 107 TSC2 variants (32%) considered as probably neutral using a transfection-based immunoblot assay (PMID: 21309039). Above all, TSC2 mutations, rather than TSC1, occupied a higher status in TSC disease. Thus we focus on TSC2 deficient disease model. To make the background more clearly, we have added a brief comparison of TSC2 and TSC1 in the introduction of the manuscript.

Question 5: Scale bars should be added to all microscopic pictures.

Response 5: According to your suggestions, we have supplemented the scale bars in microscopic pictures (Fig. 1E, Fig. 2A, Fig. 4A, Fig. 5D, Fig. 5E and Fig. S1F). And the specific information was described in figure legends.

Question 6: Control used for in vitro and in vivo drug treatments should be disclosed – DMSO, PBS, etc. Also formulation of the drugs should be clear.

Response 6: Thanks for your advice. For the in vitro experiments, Pyrotinib was dissolved to 50 mM in DMSO and then diluted to working concentration in medium. For animal experiments, the suspension was directly prepared with normal saline and administered to mice by gavage. And the specific information of drugs described in the materials and methods 2.1 and 2.11.

Question 7: Figure 5 – absolute tumor volume numbers and radiance should be plotted over time. It is quite concerning that tumors to not appear to grow over time and at the moment of treatment start (10 days) pyrotinib group showed already reduced tumors. If available, images of all 5 mice/group should be shown. If some mice were excluded it needs to be clearly stated and argued.

Response 7: Thanks for your advices. And we seem not explained clearly in the manuscript. To better compare the difference of tumor growth between the administrated and control group, we normalized the group of Pyrotinib to control group at each time point in panels (Fig. 5A and 5B). For the number of animal in the experiment, the mice were first randomized in groups containing five animals each. After 10 days, tumor-bearing mice with approximately the same tumor volume and fluorescence were randomly divided into two groups (four mice in each group), excluding the mice with unsuitable tumor volume. And later in the process of experiments, one of the mice in the Pyrotinib group with tumor disappeared after the drug treatment. Thus we finally showed three mice in each group to objectively analyze the data, and the results were presented in our manuscript (Fig. 5).

Question 8: Figure 1C- quantification of the CDK2 and CDK4 seems to not match the western blots. Possible mistake?

Response 8: Thanks for your advices. Quantification of CDK2 and CDK4 in Fig. 1C did not looked like the trend of the strip itself. So we have re-quantified the strips, the changes of CDK2 and CDK4 normalized to β-actin were the same as before, and the difference of protein expression was shown in the Fig. 1C.

Question 9: Fig1F – in the figure it is written 621-101 cell line, but the legend states it is LAM-derived cell line.

Response 9: Thanks for your correction. In Fig. S1F, 621-101 cell line was LAM-derived cell line with TSC2-deficient. Thus it is more exacted to change “621-101” to “TSC2-/- LAM-derived cells”

Question 10: Figure 5D – low resolution of the images should be improved.

Response 10: Thanks for your advices. Fig. 5D actually shows the original TUNEL assay picture, but its poor resolution might be due to the wrong way that the pictures inserted into Word. The folder of the original picture has been attached.

Question 11: Figure2D, Figure3B,C,E, etc – student t-test in not suitable for multiple comparisons, 1-way anova with multiple comparison correction should be used.

Response 11: Thanks for your correction. We have corrected our statistical analysis methods and “two-way ANOVA was used for multiple-group comparisons” was added to the materials and methods. In addition, multiple comparison algorithm and the group replicates number also corrected and added in the corresponding Figure legends.

Question 12: Figure legends should clearly state which statistical test was used and number of biological/technical replicates should be disclosed.

Response 12: Thanks for your recommendation. As suggested, statistical test and number of biological/technical replicates were disclosed in Figure legends.

Question 13: Lines 227-228 – overstatement based on only one experiment.

Response 13: Thanks for your correction. As suggested, we have changed “the microenvironment of TSC2-deficient tumor cell induced high expression of CD24 in macrophages to promote autoimmune escape” to “the microenvironment of TSC2-deficient tumor cell might promoted high expression of CD24 in macrophages to induce autoimmune escape”.

Minor points:

Question 1: Citation for Cryptotanshinone should be added.

Response 1: Thanks for your advices. As suggested, we have cited relevant references of Cryptotanshinone as a phosphor-STAT3 inhibitor in result 3.3.

Question 2: Line 352 – obvious, not obviously.

Response 2: Thanks for your correction. As suggested, “obviously” was corrected to “obvious”.

Question 3: Pyrotinib writing small vs capital needs to be unified.

Response 3: Thanks for your correction. As suggested, all “pyrotinib” changed as “Pyrotinib” in the text.

Question 4: Fig1A consistent labeling TSC22+/+ and TSC2-/- would make understanding easier.

Response 4: Thanks for your suggestion. As suggested, MEF-derived cells labelled as TSC2+/+ and TSC2-/- in Fig. 1A.

Question 5: Bioluminescence imaging – should be stated which machine, if luciferin was injected, if yes which one and how much.

Response 5: Thanks for your suggestion. As suggested, name and model of the machine used for the animal imaging system and the dosage of luciferin were clearly recorded in the materials and methods 2.11.

Question 6: FigS1A – how exactly cell death is defined – proportion of PI+-cells?

Response 6: Thanks for your recommendation. Cell death was defined that percentage of PI+-cell relative to the total number of cells, which was detected by crystal violet staining. And it was stated in the materials and methods 2.2.

Question 7: Line 303 – “expressed” is a wrong term.

Response 7: Thanks for your correction. “expressed” has been changed as “presented” to clearly explain result.

Question 8: For apoptosis, cell cycle, nuclear and cytoplasm fractioning kits exact kits names and catalog numbers should be stated.

Response 8: Thanks for your suggestion. As suggested, name and catalog numbers of apoptosis, cell cycle, nuclear and cytoplasm fractioning kits exact kits were added to the materials and methods 2.8 and 2.10.

Question 9: Antibodies (and all other reagents) catalog numbers should be disclosed.

Response 9: Thanks for your suggestion. As suggested, catalog numbers of all reagents were disclosed in the materials and methods.

Question 10: Material and methods – it should be disclosed which alcohol was used.

Response 10: Thanks for your advices. As suggested, the concentration of gradient alcohol used in IHC has been clearly elaborated in the materials and methods 2.9.

Question 11: Catalog number, dilution missing for CDK2, CDK4, siglec-10, cl PARP antibody.

Response 11: Thanks for your correction. Catalog number and dilution of CDK2, CDK4, siglec-10, cl-PARP and HER2 were explained in the materials and methods 2.4.

Question 12: Lines 108,109 “Phosphorylated proteins were quantified by their total proteins..” – meaning unclear.

Response 12: Thanks for your correction. It may be confusing that “Phosphorylated proteins were quantified by their total proteins..” To make the description more clearly, we have changed it as “Phosphorylated proteins were normalized to their total proteins, nuclear and plasma proteins normalized to Lamin B1 and α-tubulin, and other proteins normalized to β-actin, respectively. All protein content was expressed in relative units in comparison with control samples loaded on each gel.”

Question 13: Line 366 - pyrotinib typo.

Response 13: Thanks for your correction. It has been corrected as “Pyrotinib”.

Reviewer 2 Report

This work entitled “Pyrotinib Targeted EGFR-STAT3/CD24 Loop-2 mediated Cell Viability in TSC ” by Han et al., They have reported that pyrotinib specifically targeted TSC2-deficient cells, inhibited tumor cell viability and induced apoptosis through EGFR-STAT3/CD24 Loop in vivo and in vitro. In the present study, they studied the mechanisms by which pyrotinib targets TSC2-deficient tumor cells as well as the interaction between EGFR and CD24. It is professionally written, and well presented, but there are some minor comments:-

(1). Most importantly, in this study, the author didn’t mention the reason behind using the Mouse embryonic fibroblasts (MEFs) cell line.

(2). Authors also didn’t quantify the expression of ki-67, CD24, and CDK2 done by immunohistochemistry.

(3). Regarding the western blot images, in some cases band is very big and dark and seems like is over-saturated. According to a new standard of western blot images, bands shouldn’t be oversaturated.

Author Response

Dear Reviewer,

Thank you for considering our revised manuscript “Pyrotinib Targeted EGFR-STAT3/CD24 Loop-mediated Cell Viability in TSC” for the Journal of cells.

We addressed all comments to you,  and revised text and figures have corrected with another reviewer, and labelled in the manuscript. Thanks for your consideration of our work!

Question 1: Most importantly, in this study, the author didn’t mention the reason behind using the Mouse embryonic fibroblasts (MEFs) cell line.

Response 1: Thanks for your comments. For TSC research, there are three main sources of TSC2-/- and TSC2+/+ cells that have been widely accepted: Mouse embryonic fibroblasts (MEFs), LAM patient-derived cells and Eker rat uterine leiomyoma-derived (ELT3) cells. In the study of Pyrotinib selectively targeting TSC2-null cells, compared with LAM and ELT3 cells, TSC2 deficient MEF cells was more sensitive to Pyrotinib without affecting the viability of TSC2-added cells. Therefore, MEF was identified as the main disease model in our study. Related explanation has been added to result 3.1.

Question 2: Authors also didn’t quantify the expression of ki-67, CD24, and CDK2 done by immunohistochemistry.

Response 2: Thanks for your suggestion. As suggested, we have semi-quantified the expression of ki67, CD24 and CD2 using Image J software. And the quantization panel was attached in the supplementary data (Fig S5).

Question 3: Regarding the western blot images, in some cases band is very big and dark and seems like is over-saturated. According to a new standard of western blot images, bands shouldn’t be oversaturated.

Response 3: Thanks for your suggestions. We apologized that some of the bands seemed dark or over-saturated, but it might due to the size of image. So we fixed the aspect ratio of the stripes and adjusted the picture to make it more reasonable. Moreover, for some images with over-saturated, we have changed some pictures under the same experimental conditions (Fig. 3A).

Round 2

Reviewer 1 Report

Thank you for adressing the comments.